# Optimizing Calibration for a Capacitance-Based Void Fraction Sensor with Asymmetric Electrodes under Horizontal Flow in a Smoothed Circular Macro-Tube

**DOI:** 10.3390/s22093511

**Published:** 2022-05-05

**Authors:** Moojoong Kim, Kanta Komeda, Jongsoo Jeong, Mizuki Oinuma, Tetsuya Sato, Kiyoshi Saito

**Affiliations:** 1Research Institute for Science and Engineering, Waseda University, 3-4-1 Okubo, Shinjuku-ku, Tokyo 169-8555, Japan; jeong@aoni.waseda.jp; 2Department of Applied Mechanics and Aerospace Engineering, Waseda University, 3-4-1 Okubo, Shinjuku-ku, Tokyo 169-8555, Japan; komeda@akane.waseda.jp (K.K.); mi3zu2ki2-action@fuji.waseda.jp (M.O.); sato.tetsuya@waseda.jp (T.S.); saito@waseda.jp (K.S.)

**Keywords:** void fraction, capacitance sensor, calibration method, real-time measurement, quick-closing valve, R32 refrigerant

## Abstract

In this study, a technique that uses a capacitance sensor with an asymmetric electrode to measure the void fraction of a refrigerant was developed. It is known that the void fraction and flow pattern affect the measured capacitance. Therefore, the relationship between the void fraction and capacitance is not linear; hence, a calibration method for obtaining accurate measurements is necessary. A calibration method was designed in this study based on repeated capacitance measurements and the bimodal temporal distribution to calibrate the atypical and repetitive flow patterns of slug flow and its transition to the intermittent flow regime. The calibration method also considers the weighted-average relation for the gradual transition of the intermittent to annular flow pattern according to the change from low to high quality. The proposed method was experimentally analyzed under the conditions of R32 refrigerant, a tube inner diameter of 7.1 mm, saturation temperature of 25 °C, mass flux of 100–400 kg m^−2^ s^−1^, and vapor quality of 0.025–0.900, and it was validated using a quick-closing valve (QCV) system under identical conditions. A relative error of 2.99% was obtained for the entire system, indicating good agreement between the proposed and QCV-based methods.

## 1. Introduction

The void fraction is a dimensionless number that represents the fraction of the spatial or temporal domain that is occupied by the gas phase in a two-phase flow. In two-phase flow analysis, the void fraction is an essential parameter related to the flow and heat-transfer characteristics. In particular, when applied to HVAC systems and their simulations, the value of the void fraction is closely related to predicting the refrigerant charge amount assumption inside the system and estimating the performance of the system [1]. Because the refrigerant charge is directly related to the void fraction distribution in the two-phase flow region inside the system, an accurate prediction of the void fraction is required, as well as an accurate measurement of the void fraction [2]. Thus, as shown in Table 1, many researchers in various engineering fields have attempted to accurately measure and predict the void fraction in the two-phase flow region for a variety of refrigerants, heat-transfer modes, and tube shapes and sizes.

To date, various techniques for void fraction measurement have been proposed. The conventional quick-closing valve (QCV) system is one of the most well-known measurement systems. It has been applied by many researchers to measure the void fractions of refrigerants. As shown in Table 1, many predictions of the void fraction of the refrigerant have been verified based on the results of QCV-system-based measurements [3,4,5,6,7,8,9,10,11,12,13,14,15,16,17,18,19,20,21,22,23]. The QCV technique is a mechanical method that entails the use of a relatively simple configuration (two valves and a bypass) to physically lock the test section, recover the liquid inside, and measure the weight (a more detailed description is provided in Section 3). As can be expected from a simple configuration, the QCV technique has the advantages of intuitive measurement results and simple data reduction. In addition, it is reasonable to expect that the measured data accumulated from many researchers are reliable. However, there are possibilities for relatively large errors because of (1) spatial constraints, which become relevant under conditions in which only the two-phase flow between two valves is included in the measurement section, (2) time constraints related to flow cessation, and (3) the constraints related to contact methods.

The optical method entails the use of a high-speed camera and void fraction measurement to analyze two-phase flow. Although the error factor can be kept relatively low because it is a non-contact measurement method, a high-cost, high-speed camera is required, and a relatively complex data reduction technique is required to determine the void fraction. Wojtan et al. [7,16] measured the void fraction via visualization; they employed a high-speed camera, a laser, and image analysis techniques. The results obtained via this method were found to be in good agreement with the Rouhani–Axelsson drift flux model [24] modified by Steiner [25]. However, the applied image analysis technique only yielded reasonably accurate results for the stratified flow pattern (conditions: low mass flux and large diameter). Keinath et al. [21] derived the void fraction from the reconstruction results for a two-phase flow, which were obtained by utilizing a two-way orthogonal high-speed camera as the measurement tool. Then, they derived the spatial distribution of the void fraction in the axial direction of the test section. However, the uncertainty of the applied technique was relatively high, and the results tended to not agree well with the existing void fraction correlation.

Alternatively, the capacitance technique is a measurement method that exploits the electrostatic properties of the two-phase flow between two electrodes [18,19,22,26,27]; a more detailed description is provided in Section 3. Given that the technique utilizes a relatively simple configuration, enables real-time measurements, and has the characteristics of a non-contact method, it has recently begun receiving attention. In particular, the characteristic of real-time measurement of a void fraction is useful in terms of evaluation of recent heat exchangers with a complex refrigerant flow distribution structure to improve performance. The electrostatic parameter applied in this technique is the dielectric constant, and the capacitance measured through the two electrodes varies according to the type of fluid, void fraction, and flow pattern. Here, the relationship between the capacitance and void fraction (i.e., the *C–**⍺* relation) is not linear, and the *C–**⍺* relation changes according to the flow pattern. Therefore, there is a dependency on the flow pattern. Specifically, the flow pattern is dependent on the mass flux, temperature, flow rate, and refrigerant under two-phase flow. Thus, the process of using a capacitance sensor to determine the void fraction requires that the *C–**⍺* relation be quantified according to the pattern of the two-phase flow, i.e., through the use of a calibration method.

Interestingly, studies that have entailed investigating the dependency on the flow pattern have focused on maximizing or minimizing this dependency. Without focusing on the void fraction, Canière et al. [28] maximized the dependency of a capacitance sensor on the flow pattern to fabricate a sensor that could detect the pattern of two-phase flow. Conversely, de Kerpel et al. [19] measured the void fraction and performed finite element method (FEM)-based simulations to calibrate slug, intermittent, and annular flows by applying a concept of flow pattern dependency minimization. The results that they obtained were in good agreement with the Steiner version of the Rouhani–Axelsson drift flux model. However, their method was not directly compared to other measurement techniques under the same experimental conditions.

As can be understood from these studies, the use of capacitance sensors to measure the void fractions of refrigerants still has its challenges. For example, a capacitance sensor shows different results for the void fraction depending on the method used to calibrate the flow pattern, even though it can provide excellent characteristics, such as non-contact and real-time measurements [29,30]. Therefore, the reliability of the capacitance sensor should be determined using a proper calibration method. In addition, this calibration method should be validated using reliable conventional techniques, and the method should consider the dependency on the flow pattern and its transition.

This study aimed to validate a capacitance method for void fraction measurement by applying a capacitance sensor [31] with low sensitivity to flow patterns. To achieve this aim, a calibration method was established based on the characteristics of various flow patterns that were visualized by using a capacitance sensor. The observed flow pattern was reflected in the results of a finite element method (FEM) analysis to derive the *C–**⍺* relation for the capacitance sensor. The proposed calibration method was validated against the QCV method, which was simultaneously applied to the horizontal flow of the smoothed circular macro-tube of the R32 refrigerant under the following conditions: 25 °C saturation temperature, a 100–400 kg m^−2^ s^−1^ mass flux range, and *x* = 0.025–0.900 vapor quality.

## 2. Experimental Apparatus

Figure 1 shows a schematic of the experimental apparatus used to measure the void fraction. The refrigerant loop was made up of a copper tube with an inner diameter of 7.1 mm; it flowed from the receiver tank to the preheater through the sub-cooler by means of a magnetic geared pump (GA series, MICROPUMP) to prevent oil mixing and the pulsation caused by using a compressor and centrifugal pump. A typical air conditioner uses a compressor to circulate the refrigerant. However, the compressor was lubricated for smooth mechanical operation, and the lubricant oil was mixed into the refrigerant. Since mixed lubricant oil can affect the characteristics of the refrigerant, it should be excluded from element studies such as this study. If a centrifugal pump is used to circulate the refrigerant, pulsation of the flow may be induced. The above oil mixing and pulsation problems can be error factors in experiments. The refrigerant in the supercooled state was heated by the preheater and converted into its two-phase state with a specific inlet vapor quality. The refrigerant in the two-phase state was returned to the refrigerant tank and passed through the test section (length: 1.2 m, inner diameter: 7.1 mm); the gas phase of the refrigerant was then condensed in the after-condenser. To prevent heat loss to the surrounding environment, the entire experimental apparatus was insulated by using 15-mm-thick AEROFLEX insulation. The flow rate of the refrigerant was controlled by adjusting the revolutions per minute of the magnetic geared pump and was measured by using a flow meter (OVAL, ALTImass II type U) with an accuracy of ±0.05%. The inlet vapor quality was adjusted by using an electric preheater with a capacity of 5.6 kW; the power consumption of the heater was measured by using a power analyzer (YOKOGAWA, WT332E) with ±0.1% accuracy. The saturation temperature and pressure at the test section were controlled by using a separate external temperature control system to regulate the cooling capacity of the condenser. Simultaneously, the pressure was measured by using a pressure transducer (YOKOGAWA, EJA430J) with ±0.1% full-scale accuracy, and the temperature was measured by using a PT100 RTD sensor (SAKAGUCHI, Class A) with ±0.15 °C accuracy. The uncertainty of the experimental apparatus for the inlet vapor quality was ±2.12% (*x* = 0.5, 25 °C, 250 kg m^−2^ s^−1^). A high-speed camera (IDT, NR4-ANM1) was applied to the visualization section mounted downstream of the test section to observe the flow pattern under all inlet vapor quality conditions. The high-speed camera system operated under the conditions of 1016 × 1016 pixels and a sensor of 1 million pixels; it was also equipped with a 50 mm f/2.8 macro lens (Nikon), and 2 s of flow was captured under the conditions of 5000 fps and 139 µs of exposure time.

Three types of mass flux conditions were selected under adiabatic conditions for application to the experimental apparatus based on the Wojtan–Ursenbacher–Thome flow pattern map results (Figure 2) [32]. Various flow patterns were implemented under the three mass flow conditions to establish the calibration method: slug flow, stratified flow, stratified–wavy flow, intermittent flow, annular flow, and the transitions between flow patterns. To minimize heat loss, the saturation temperature was set to 25 °C. R32 was employed as the working refrigerant, and the applied saturation pressure was 1.6896 MPa. The inlet vapor quality was varied between 0 and 0.900 at an interval of 0.1. However, generally, because the low-quality range (*x* = 0–0.2) is a region associated with rapid transitions in the void fraction and flow pattern, the inlet vapor quality in this range was varied at an interval of 0.025 during the measurement. All conditions were applied in three independent experiments to confirm repeatability. Table 2 summarizes the experimental conditions applied in this study.

## 3. Methodology

Void fraction measurement is necessary for the accurate prediction and estimation of the heat-transfer and hydrodynamic parameters of a two-phase flow [32]. Thus, an understanding of each void fraction measurement technique is very important for studies on two-phase flow characteristics. In this section, the QCV and capacitance techniques applied in this study are described. Additionally, the measurement apparatus, uncertainty, and data reduction capability of each technique are described in detail. Lastly, a calibration method that enables accurate void fraction measurement via the capacitance technique is proposed.

### 3.1. Measurement Apparatus for the QCV Technique

Figure 3 shows a detailed schematic of the test section. The test section comprised a 1.2-m-long and 7.1-mm-inner-diameter copper tube and was designed to maintain a fully developed flow through identical 1-m-diameter upstream and downstream tubes. The QCV technique entails the application of three solenoid valves, i.e., A, B, and D (normally close), a bypass line, and a QCV recovery line located downstream of the test section. Here, the QCV method was only limited to the two-phase fluid present in a volume of approximately 50 mL between solenoid valves A and B. In general, the longer the distance between the two valves and the larger the internal volume, the smaller the measurement error of the QCV method; however, in this study, the distance was limited to 1.2 m owing to space constraints. The pressure (*P_rc_*) and temperature (*T_rc_*) of the QCV recovery line were measured to increase the accuracy of the void fraction measurement technique. The volumes of the test section (*V_ts_*) and QCV recovery line (*V_rc_*) are essential parameters for the reduction of the void fraction data. Here, the volume of the QCV recovery line (*V_rc_*) was occupied by the saturation gas phase of the refrigerant during the QCV measurement process; its mass should be considered in the data reduction of the void fraction. These two volumes were measured by performing a single-phase flow experiment and had uncertainties of 1.1% and 1.05%, respectively.

After being passed through the preheater, the subcooled refrigerant was transformed into a two-phase flow with a predetermined vapor quality. At this time, solenoid valves A and B were opened, and D was closed. The two-phase refrigerant flowed into the test section along the blue arrow. When the incoming two-phase flow reached the steady state, solenoid valves A and B were closed, and D was opened; the two-phase refrigerant flowed into the bypass line along the red arrow. All of the solenoid valves were opened/closed within 3 ms. Although valve opening/closing is fast enough, the contact method has limitations. In atypical flows, such as slug flow patterns, the slug frequency between two solenoid valves varies depending on the valve opening/closing time. The amount of refrigerant in the tube varies depending on the trapped slug frequency, and this is reflected as an error in the void fraction measurement when the QCV method is used. Before the steady state was reached, the entire QCV recovery line and recovery vessel were maintained in a vacuum, and the recovery vessel was cooled to −10 °C. When valve C, which was located upstream of the QCV recovery line, was opened, the refrigerant was evaporated, converted into a superheated state, and condensed as it traveled toward the recovery vessel. For a short period of time, the test section was heated to ensure sufficient evaporation. However, sufficient condensing time is required even if a short heating time is considered, and the longer the recovery time through condensation, the smaller the error. In this study, we performed condensate recovery for approximately 20 min per measurement point owing to time constraints. After the condensed water was removed from the surface, the sufficiently condensed recovery vessel was weighed using a precision scale (VIBRA, AJ2-2200). The uncertainty in the QCV-based void fraction measurement technique was 1.07%.

### 3.2. Data Reduction for the QCV Technique

Mass flux (G) was calculated as described by Equation (1).
(1)$$G=4\frac{\dot{m}}{\pi D^{2}}$$
where $\dot{m}$ is the measured mass flow rate and D is the inner diameter of the test section.

The inlet vapor quality ($x_{ts,\;in}$), which is a common parameter of the QCV and capacitance techniques, can be calculated by using Equations (2) and (3).
(2)$$x_{ts,\;in}=\frac{i_{ts,in}-{i_{f}|}_{T_{ts,in}}}{{i_{fg}|}_{T_{ts,in}}}$$
where ${i_{f}|}_{T_{ts,in}}$ and ${i_{fg}|}_{T_{ts,in}}$ are the saturated liquid specific enthalpy and specific latent enthalpy of the test section, respectively. Here, $i_{ts,in}$ is the specific enthalpy of the two-phase flow at the entrance to the test section; it can be calculated by using Equation (3).
(3)$$i_{ts,in}=i_{ph,in}+\frac{Q_{ph}}{\dot{m}\;}$$
where $i_{ph,in}$ is the specific enthalpy of the subcooled liquid refrigerant at the inlet of the preheater and $Q_{ph}$ is the measured amount of power supplied to the preheater, which considers heat loss in the preheater section, as follows in Equation (4).
(4)$$Q_{ph}=Q_{ph,in}+\dot{m}C_{p,R}\left(T_{ts,in}-T_{ph,in}\right)$$
where $Q_{ph,in}$ is the actual power consumption at the preheater, $C_{p,R}$ is the specific heat capacity of the refrigerant, and $T_{ts,in}$ and $T_{ph,in}$ are the inlet temperatures of the test section and preheater section, respectively. The heat loss measurement was performed at the same temperature as the saturation temperature of the void fraction measurement using a single-phase R32 refrigerant.

The void fraction, as determined via the QCV technique ($\alpha_{\mathrm{QCV}}$), is defined as follows:(5)$$\alpha_{\mathrm{QCV}}=\frac{M_{R,ts}+\rho_{g.rc}\left(V_{ts}+V_{rc}\right)-\rho_{f.ts}V_{ts}}{\left(\rho_{g.ts}-\rho_{f.ts}\right)V_{ts}}$$

Here, $M_{R,ts}$ is the mass of the refrigerant recovered from the test section, $\rho_{f.ts}$ and $\rho_{g.ts}$ are the density of the each phase of refrigerant in the test section, and $\rho_{g.rc}$ is the density of the gas phase of the refrigerant in the QCV line.

### 3.3. Measurement Apparatus for the Capacitance Technique

As shown in Figure 3, a capacitance sensor was mounted at the center of the test section. The capacitance sensor had an upstream calming section longer than 50 diameters (50D), which was a 0.45 m length after the solenoid valve (B). This calming section ensured a fully developed flow of two-phase flow. To measure the void fraction of the refrigerant, the sensitivity to factors affecting its flow pattern must be relatively low. More specifically, a low-sensitivity flow pattern refers to a flow pattern that is not sensitive to macroscopic changes, e.g., the difference between stratified flow and annular flow, but is sensitive to microscopic changes, e.g., small differences in the process of gradual transition from stratified flow to annular flow. The sensor used in this study was designed in accordance with a previously reported design methodology [31]. Thus, we present an overall outline of the sensor in Figure 4. Figure 4 shows the exact shape and specifications of a capacitance sensor that was designed based on the results of FEM-based optimization. Polyether ether ketone (PEEK) material was used for the piping to ensure that the measurements took into account any changes in the electric field. The capacitance due to the two-phase flow between the two electrodes was measured by using a capacitance meter (HIOKI, C HiTESTER 3506-10) with a measurement error of 0.12%. The designed sensor had an error of 3.47% (stratified flow, ⍺ = 0.5, 25 °C, 250 kg m^−2^ s^−1^).

### 3.4. Data Reduction Scheme for Capacitance Method for Void Fraction Measurement

As described above in Section 1, an understanding of the C–⍺ relation is required to obtain accurate void fraction measurements using a capacitance sensor. For instance, assuming that there is a *C–**⍺* curve, as indicated by the black line in Figure 5a, the capacitance (80 fF, in Figure 5a, green dot-line) measured from the sensor and the void fraction (0.7, in Figure 5a, green dot-line) can be matched. However, the *C–**⍺* relation changes to the red or the blue line in Figure 5a depending on the flow pattern. Therefore, if the analysis and calibration of the flow pattern are insufficient, the void fraction of the two-phase fluid cannot be determined using the measured capacitance result (80 fF, in Figure 5a).

Thus, we utilized FEM analysis to derive the *C–**⍺* relation according to the flow pattern and void fraction. FEM analysis was performed using Elmer, an open-source software program, and Maxwell’s equation was used for electric field analysis (EFA). In Elmer, the Dirichlet or Neumann condition can be applied as a potential boundary condition for capacitance analysis; in this study, the Dirichlet condition was used. Potential conditions were applied on the electrode surface: 1 V at the high-potential plate and 0 V at the low-potential plate. Before evaluating the effects of various flow patterns on the *C–**⍺* relationship, for validation, the capacitance in a single phase was measured by using a dummy fluid (NOVEC7300, permittivity(ε) = 6.1) with the same properties as the R22 refrigerant (ε = 6.1); the results are shown in Table 3. The entire process of void fraction measurement, including the process applied to derive the *C–**⍺* relation, is illustrated in Figure 5.

As shown in Figure 2, the experimental range of this study included various flow patterns. Generally, the flow pattern is dependent on the flow rate and vapor quality of the two-phase flow. Here, the flow pattern type was separated into two general groups, as depicted in Figure 6; group 1: the slug to intermittent flow transition, group 2: the intermittent to annular flow transition.

Because the asymmetric capacitance sensor used in this study was not highly sensitive to changes in the flow pattern, small changes in the flow patterns minimally affected the measured capacitance. Given this, bubble, stratified, and annular flow regimes were selected to be the representative flow patterns for FEM analysis; the consequently derived *C–**⍺* curves are shown in Figure 7. Furthermore, the void fraction of each flow pattern ($\alpha_{strf}$, $\alpha_{bbl}$, $\alpha_{ann}$) can be decided from these *C–**⍺* curves.

A linear *C–**⍺* curve is also shown in Figure 7 for the purpose of comparison to the non-calibrated condition. Note that the y axis in Figure 7 shows the normalized capacitance (*C_n_*). Because the dielectric constant varies according to temperature, a standardized value that takes into account any changes in fluid temperature was applied to minimize the influence of temperature on the capacitance measurement. The normalized equation is given as Equation (6).
(6)$$C_{n}=\frac{C_{m}-{C_{g}|}_{T_{ts}}}{{{C_{l}|}_{T_{ts}}-C_{g}|}_{T_{ts}}}$$

Here, $C_{m}$, $C_{g},$ and $C_{l}$ are the capacitance valves of the measurement, single gas phase, and single liquid phase, respectively.

### 3.5. Void Fraction Measurement Methodology for the Calibrated Capacitance Sensor

When a capacitance sensor is used to perform void fraction measurements, it is critical that the applied conversion relation (*C–**⍺* relation) is suitable for the actual flow pattern. It is generally difficult to select the most suitable conversion relation, and it is necessary to ensure the reliability of the capacitance sensor. In particular, it is critical to select and apply a conversion formula for the slug flow region (i.e., where two flow patterns are repeated) and a transition region represented by the intermittent flow (i.e., where the flow pattern changes) region. Furthermore, even under the conditions of the application of a capacitance sensor with relatively low sensitivity, the *C–**⍺* relation would still differ according to the flow pattern, as shown in Figure 7. Therefore, an appropriate calibration method is absolutely essential for the accurate measurement of the void fraction.

#### 3.5.1. The Transition Region: The Asymmetric to Symmetric Flow Pattern

In this study, the calibration technique applied to the transition region was developed based on the calibration method proposed by de Kerpel et al. [19]. The corresponding calibration equation proposed by de Kerpel et al. is given as Equation (7); de Kerpel and colleagues applied this equation to the Wojtan–Ursenbacher–Thome flow pattern map [32] and Barbieri flow pattern map [33]. In addition, de Kerpel’s research is limited by not directly comparing the void fraction measured by the capacitive method based on the proposed calibration method with the void fraction from another reference method, such as the QCV method. Steiner correlation [25], a comparison target in de Kerpel’s research, is practically relatively accurate, but it has an error of about 2–7% depending on the refrigerant.
(7)$$\alpha_{intm}=\frac{\left(x-x_{IS}\right)}{\left(x_{IA}-x_{IS}\right)}\alpha_{ann}+\frac{\left(x_{IA}-x\right)}{\left(x_{IA}-x_{IS}\right)}\alpha_{slug}$$

De Kerpel et al. also reviewed and proposed a calibration technique for the void fraction of intermittent area ($\alpha_{intm}$) [19] based on the weighted-average relation of the void fraction of annular flow ($\alpha_{ann}$) and slug flow ($\alpha_{slug}$). However, the calibration was performed from the intermittent–slug interface (*x_IS_*) to the intermittent–annular interface (*x_IA_*). Such a process would inevitably lead to the formation of discontinuity points at the flow pattern interface. This is because the change in the actual flow pattern is continuous, not discontinuous, as it is represented in the flow pattern maps (Figure 2). As such, there is a gradual change from an asymmetrical flow pattern (slug flow, stratified flow) to a symmetrical flow pattern (complete annular flow, *x*≈1). Hence, this characteristic should be appropriately taken into account in the calibration method.

Taking this into consideration, the calibration method applied in this study was designed to account for the gradual change from an asymmetrical to a symmetrical flow pattern, as expressed in Equation (8). This equation is the extended version of the equation by de Kerpel et al., covering the intermittent flow, the annular flow regime, and *x* = 1 (single-gas-phase flow).
(8)$$\alpha_{intm\;\&\;ann}=\frac{\left(x-x_{IS}\right)}{\left(x_{EA}-x_{IS}\right)}\alpha_{ann}+\frac{\left(x_{IA}-x\right)}{\left(x_{EA}-x_{IS}\right)}\alpha_{strf}$$

The flow pattern was classified by using the Wojtan–Ursenbacher–Thome flow pattern map. Here, *x_EA_* is the boundary at the end of the annular flow and is equal to 1 under adiabatic conditions. However, although the Wojtan–Ursenbacher–Thome flow pattern map is consistent with most visualized results, the boundary between the slug flow and stratified–wavy flow is not clear in the low mass flux range (100 kg m^−2^ s^−1^ in this experiment). Thus, in this study, the calibration method was applied based on the visualized results at 100 kg m^−2^ s^−1^.

#### 3.5.2. Slug Flow Regime: Dual Characteristics of the Flow Pattern

In the case of the slug flow regime, the characteristics of the flow pattern and selection of the experimental representative values must be taken into consideration for the calibration method. It should be noted that the flow pattern can be indirectly observed by using a capacitance sensor [28,29]. This is because the capacitance measurement method affords a relatively high signal-to-noise ratio and fast response speed. As shown in Figure 8a, these characteristics allow the capacitance to be measured in real time, which is closely related to the void fraction. The real-time capacitance history can be changed according to the flow pattern. So, the temporal distribution of the void fraction, which is closely related to the actual flow pattern of the two-phase flow, can be derived. Figure 8b,c show the typical profiles for capacitance measured over time for the slug and stratified flow regimes. Here, the average and interquartile range (IQR) average were used for representative value analysis, and the interquartile range average was used to remove outliers of the measurement results from the analysis of each peak.

The slug flow profile in Figure 8b is bimodal. Furthermore, the slug flow contained a low-capacitance region (corresponding to stratified flow) and high-capacitance region (corresponding to bubbly flow). Alternatively, in the case of the stratified flow (Figure 8c), it can be observed that the distribution of capacitance was stable and had a single mode. Moreover, the annular flow regime profile is represented in Figure 8d; it showed stable temporal distribution. A histogram was used to allow visualization of these characteristics. Figure 8e shows the measured capacitance profiles with respect to the inlet vapor quality, which was increased under the conditions of *G* = 100 kg m^−2^ s^−1^. In the case of very low inlet vapor quality (*x* = 0.025–0.075), the bimodal distribution was the result of many repetitions of stratified flow and bubbly flow patterns. The bimodal distribution gradually changed to a single-mode distribution as the inlet vapor quality increased (*x* ≥ 0.2); this implies that the slug flow was converted into a fully stratified flow.

Obtaining the void fraction from the bimodal measured capacitance profile of the slug flow required a post-processing standard for conversion of the raw data into a void fraction. Figure 8b shows various representative values of the bimodal capacitance distribution for the slug flow. As an example, if the average values of all of the relevant capacitance data shown in Figure 8b were used, the results would be values that are not representative of the capacitance distribution. This would not facilitate the determination of the true *C*–*⍺* relationship. It is essential to accurately determine the *C*–*⍺* relationship because, in the low-inlet-quality region, the converted void fraction result significantly varies for the same capacitance.

Thus, in this study, the peak analysis method for the bimodal characteristics of the slug flow capacitance was applied in the void fraction conversion to take into account the physical phenomena of the slug flow. As an example, from the measurement results shown in Figure 8, this distribution can evidently be partitioned into multiple single-mode distributions based on the average value of all the corresponding data. It is reasonable to state that the first distribution (*fd*) on the left has the characteristics of a stratified flow and that the second distribution (*sd*) on the right has the characteristics of a bubbly flow. From this, it is possible to determine the proportions of stratified flow (*Count_fd_/Count_tot_*) and bubbly flow characteristics (*Count_sd_/Count_tot_*) with respect to the total number of measurements. Then, this information can be used to develop an equation that well reflects the actual *C–**⍺* relationship. The void fraction of the slug flow can be expressed as
(9)$$\alpha_{slug}=\frac{Count_{fd}}{Count_{tot}}\alpha_{strf}+\frac{Count_{sd}}{Count_{tot}}\alpha_{bbl}$$

## 4. Results and Discussion

In this study, we performed validations on the advantage of void fraction estimation using the QCV technique compared to that using specific correlations. Although several correlations, such as those of Smith, Steiner, and Zivi, can predict the void fraction with an acceptable level of accuracy, there is a significant difference from the actual void fraction measurement results. Thus, actual measurement data obtained through the use of a conventional, reliable technique are required to validate the new measurement method. In this study, the QCV-technique-derived void fractions were measured simultaneously with the capacitance sensor under the same experimental conditions; this allowed a direct comparison without considering the correlation accuracy.

Figure 9 shows the void fraction results that were obtained by applying the QCV technique and the capacitance method with both calibration methods. In addition, the void fraction from the Steiner correlation is also shown for comparison. However, we note that the void fraction measurement result of the R32 refrigerant with a smooth circular macro-tube was not compared with those from the Steiner correlation and discussed; this will be reported in another paper. Evidently, the linear calibration (i.e., blue points) consistently yielded results that were significantly different from the results obtained via the QCV technique. In particular, the differences in the intermittent regime were the largest, whereas the differences were relatively small for the slug, stratified, and annular flow regimes. Nevertheless, these differences were minimized by applying the proposed calibration method, as shown by the red points in Figure 9.

Evidently, the capacitance was slightly underestimated under the conditions of an intermediate inlet vapor quality range (i.e., *x* = 0.3–0.6), T = 25 °C, and G = 250 kg m^−2^ s^−1^. These underestimations are related to the QCV measurements; particularly, it is assumed that the underestimated values were related to the QCV recovery line located downstream of the test section, which is essential for QCV void fraction determination. This will be comprehensively investigated in a subsequent study. To evaluate the deviation according to the flow pattern in the void fraction measurement, the flow pattern was considered based on the Wojtan–Ursenbacher–Thome flow pattern map and the visualization results in the calibration, as shown in Figure 9. However, as mentioned before, in the case of the slug flow regime, there were some cases in which the actual two-phase flow was found to have patterns that were not included in the Wojtan–Ursenbacher–Thome map. Therefore, a calibration technique was developed based on the results of high-speed camera observations and analysis of the distribution of repeated capacitance measurements.

The results of applying the proposed calibration technique to the capacitance sensor were compared to those obtained via the QCV technique; the results are summarized in Table 4.

In Table 4, *E_A_* is the average relative error between the void fractions obtained by using the calibrated capacitance and QCV techniques. The results indicate that void fraction measurement via the capacitance method can be carried out more accurately when the applied calibration technique takes into account the slug flow characteristics and the gradual transitions of the overall flow pattern, rather than being simple linear calibration. Furthermore, it can be confirmed that the calibration method proposed in this study agreed well with the QCV void fraction measurement results under all experimental conditions, including the inlet vapor quality range, achieving *E_A_* = 2.99%, and *R*^2^ = 0.994. Thus, it can be concluded that the capacitance sensor to which the proposed calibration method is applied is a practical alternative to the QCV technique for void fraction measurements.

## 5. Conclusions

In this study, a flow-pattern-based calibration technique for capacitance sensors was developed and subsequently validated by comparing the void fraction results to those obtained via the QCV technique under the same experimental conditions. By using the QCV void fraction measurement results instead of the prediction correlations for the void fraction, the behavior of the capacitance sensor could be thoroughly investigated. The flow pattern characteristics were taken into account in the proposed calibration method upon analysis of the visualized results based on the Wojtan–Ursenbacher–Thome flow pattern map. The *C–**⍺* relationship for the representative flow pattern was derived based on the results of an FEM simulation. In the case of the slug flow, a new calibration method using two capacitance distributions of the slug flow pattern was proposed; the distributions were determined based on the analysis of repeated capacitance measurements that were averaged according to the ratio of each distribution. The calibration method for after the slug flow regime is an extended version of the method proposed by de Kerpel et al., which takes into consideration the occurrence of gradual transitions between asymmetry and symmetry in horizontal flow. Thus, the void fraction results obtained via the capacitance measurement method with the proposed calibration technique were in good agreement with those obtained via the QCV method, with *E_A_* = 2.99% and *R*^2^ = 0.994 for the entire test section. These results demonstrate the potential of the proposed optimally calibrated capacitance method as a replacement for the QCV measurement method. However, the generalizability of the proposed calibration technique, particularly for application to various refrigerants, pipe diameters, temperatures, etc., must be verified to establish a reliable capacitance method.

## Figures and Tables

**Figure 1 sensors-22-03511-f001:**
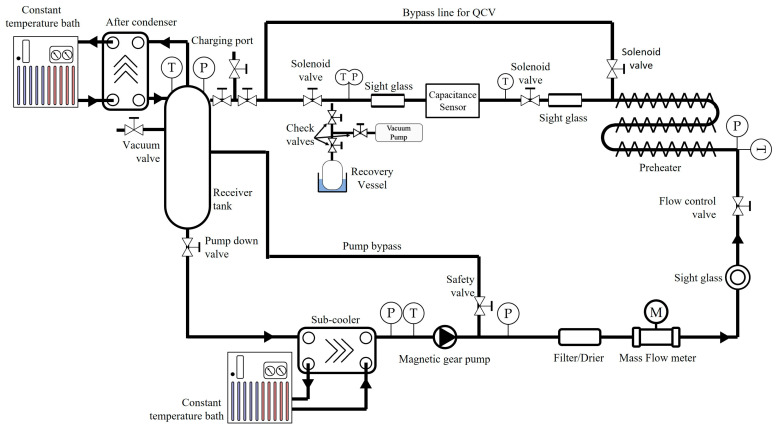
Schematic diagram of the experimental apparatus.

**Figure 2 sensors-22-03511-f002:**
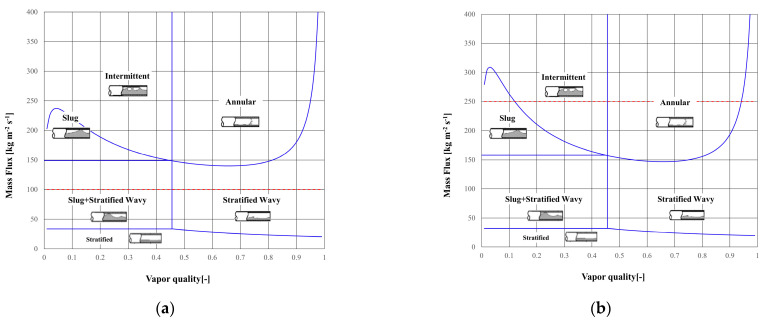
Wojtan–Ursenbacher–Thome flow pattern map for each mass flux (red dot-line) with R32 and T_sat_ = 25 °C; (**a**) 100 kg m^−2^ s^−1^; (**b**) 250 kg m^−2^ s^−1^; (**c**) 400 kg m^−2^ s^−1^.

**Figure 3 sensors-22-03511-f003:**
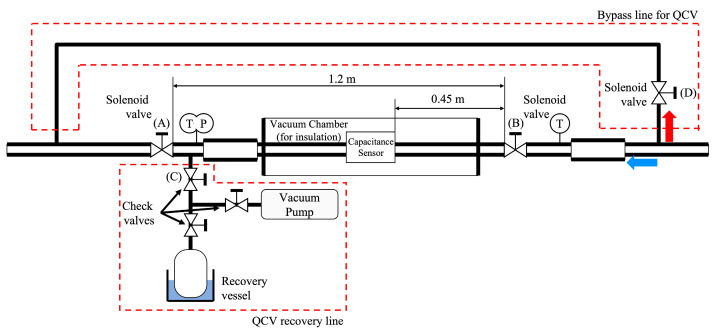
Detailed schematic of the test section.

**Figure 4 sensors-22-03511-f004:**
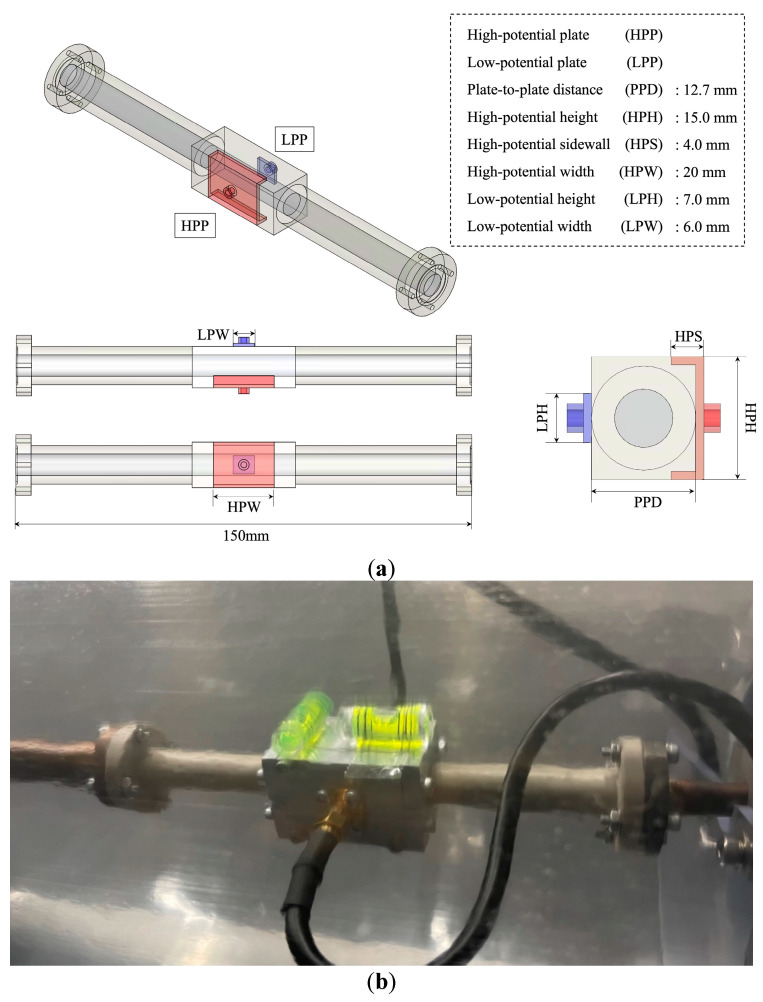
Capacitance sensor optimized for R32; inner diameter: 7.1 mm; (**a**) design parameters and the geometry; (**b**) mounted capacitance sensor inside the vacuum insulator.

**Figure 5 sensors-22-03511-f005:**
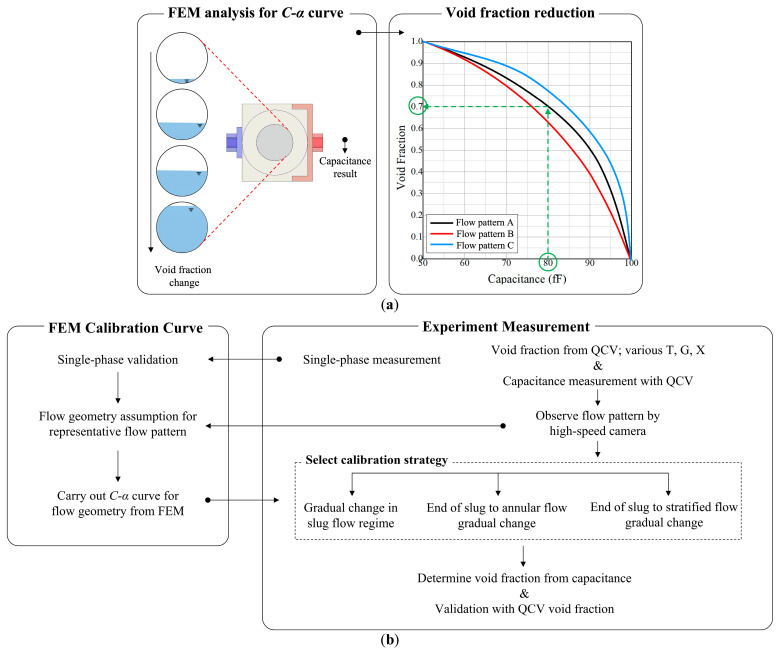
The entire void fraction measurement process for the capacitance sensor; (**a**) FEM analysis and void fraction data reduction from measured capacitance; (**b**) proposed calibration process.

**Figure 6 sensors-22-03511-f006:**
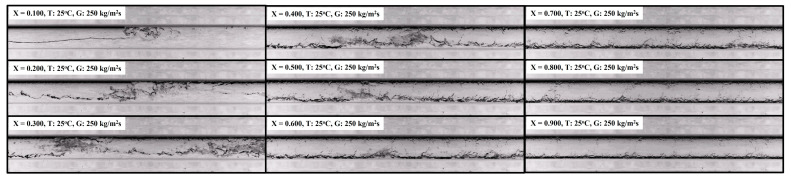
Transition patterns of the R32 flow; 25 °C, 250 kg m^−2^ s^−1^; group 1: the slug to intermittent flow transition, group 2: the intermittent to annular flow transition.

**Figure 7 sensors-22-03511-f007:**
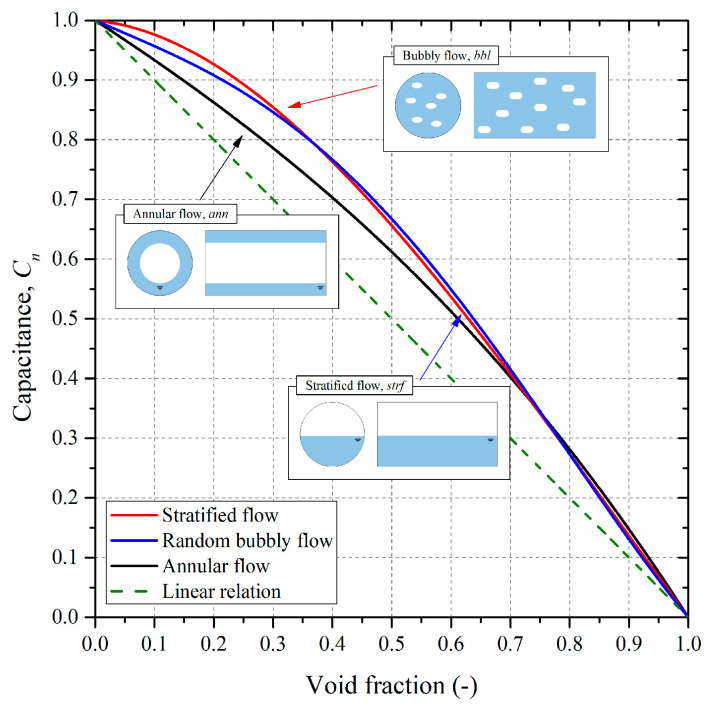
*C–**⍺* curves for each representative flow pattern analyzed using FEM analysis.

**Figure 8 sensors-22-03511-f008:**
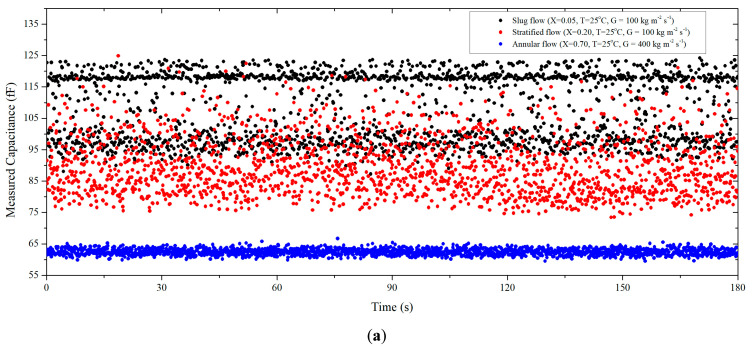
Typical distributions of capacitance measurement results for the R32 slug and stratified flow; (**a**) typical real-time capacitance histories of each flow pattern; (**b**) slug flow, x=0.05, *T* = 25 °C, *G* = 100 kg m^−2^ s^−1^; (**c**) stratified flow, x=0.200, *T* = 25 °C, *G* = 100 kg m^−2^ s^−1^; (**d**) annular flow, x=0.700, *T* = 25 °C, *G* = 400 kg m^−2^ s^−1^; (**e**) measured capacitance profiles with respect to inlet vapor quality, *T* = 25 °C, *G* = 100 kg m^−2^ s^−1^.

**Figure 9 sensors-22-03511-f009:**
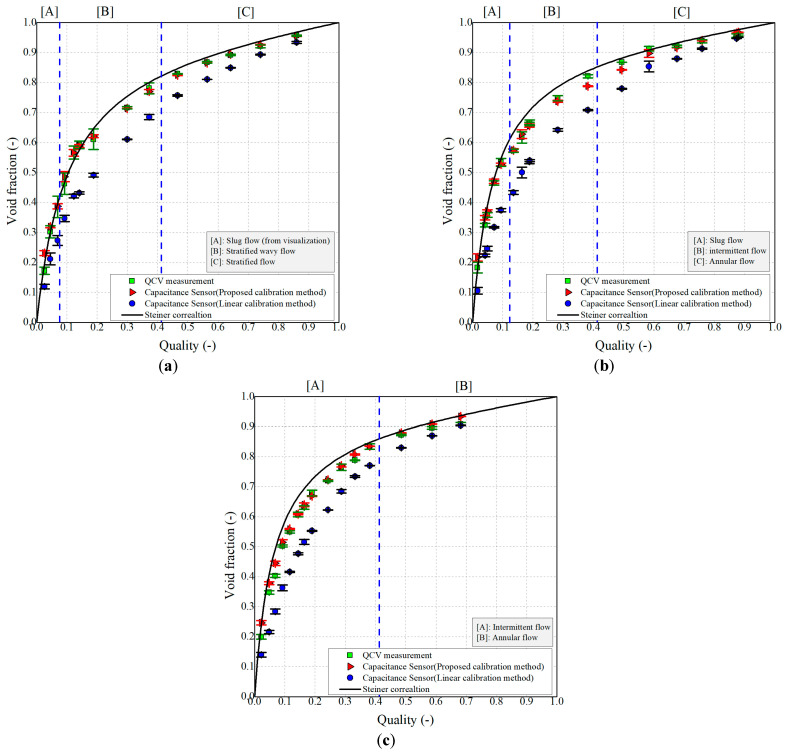
Comparison between the QCV-technique-derived void fraction results and those obtained via the capacitance method with the proposed and linear calibration; (**a**) R32, T = 25 °C, G = 100 kg m^−2^ s^−1^; (**b**) R32, T = 25 °C, G = 250 kg m^−2^ s^−1^; (**c**) R32, T = 25 °C, G = 400 kg m^−2^ s^−1^.

**Table 1 sensors-22-03511-t001:** Summary of reported void fraction measurement results for refrigerants.

Reference	Size (mm)/Shape /Orientation *	Refrigerant	Mass Flux (kg m^−2^ s^−1^)	T_sat_ (°C)	Heat-Transfer Mode	Quality	Void Fraction	Method *
Li et al. [3]	0.643/MP/H	R32	50–300	30	Evaporating	0.02–0.8	0.2–1	Optical
Pabon et al. [4]	4.8/SC/H	R1234yf, R134a	180, 280	15, 25	Adiabatic	0.1–1	0.7–1	QCV
Spencer [5]	2.99, 4.56/SC/H	R1234yf, R134a, R290	1.0–2.5 (g s^−1^)	7	Evaporating	0.1–0.8	0.6–1	QCV
Koyama et al. [6]	7.52/SC/H	R134a	125–250	31, 46	Adiabatic	0.01–0.96	0.32–0.95	QCV
Wojtan et al. [7]	8/SC/H	R22	100–150	5	Evaporating	0.05–0.22	0.61–0.78	Optical
Graham et al. [8]	7.04/SC/H	R134a, R410A	75–450	35	Condensing	0.05–0.89	0.31–0.96	QCV
Kopke et al. [9]	6.04/SC/H	R134a, R410A	75–300	35	Adiabatic Evaporating Condensing	0.01–0.45	0.33–0.88	QCV
Sacks [10]	9.6/SC/H	R11, R12, R22	82–902	25, 32	Adiabatic Condensing	0–0.98	0.28–1	QCV
Hashizume [11]	10/SC/H	R12, R22	88.5–353.9	20, 39, 50	Adiabatic	0.08–0.91	0.35–0.98	QCV
Wilson et al. [12]	6.12/SC/H	R134a, R410A	75–500	5	Adiabatic Evaporating	0.1–0.9	0.64–0.99	QCV
Yashar et al. [13]	4.26/SC/H	R134a, R410A	200–700	5	Adiabatic Evaporating	0.05–0.84	0.57–0.98	QCV
Tran et al. [14]	8.28/SC/H	R134a, R410A	53.7–355.7	5	Adiabatic	0.05–0.7	0.57–0.98	QCV
Niño [15]	1.54, 1.02/MP/H	R134a, R410A	100–300	10, 20	Adiabatic	0.05–0.71	0.60–0.99	QCV
Wojtan et al. [16]	13.6/SC/H	R22, R410A	70–300	5	Evaporating	0–0.95	0.12–0.99	Optical
Revellin et al. [17]	0.5/SC/H	R134a	1000	30	Evaporating	0.01–0.10	0.28–0.72	Optical
Shedd [18]	2.92, 1.19, 0.508/SC/H	R410A	200–850	50	Condensing	0.05–0.91	0.23–0.99	Capacitance
de Kerpel et al. [19]	8/SC/H	R134a, R410A	200–500	15	Adiabatic	0.02–0.97	0.29–1	Capacitance
Srisomba et al. [20]	8/SC/H	R134A	644–1455	23–30	Adiabatic	0–0.8	0.18–1	QCV, Optical
Keinath et al. [21]	0.508, 1, 3/SC/H	R404A	200–800	30–60	Condensing	0.05–0.95	0.25–0.75	Optical
Portillo et al. [22]	3/SC/H	R410A	200–400	30–50	Adiabatic	0–1	0–1	Capacitance
Wilson et al. [23]	7.78, 6.37, 4.40, 1.84/FT/H	R134a, R410A	65–900	35	Condensing	0.04–0.83	0.41–0.93	QCV
Summary	Tube sizes: 0.5–10 mm MP: 2 cases SC: 18 cases FT: 1 case	R11: 1 case R134a: 13 cases R12: 2 cases R290: 1 case R22: 3 cases R410A: 11 cases R32:1 case R1234yf: 2 cases	Mass flux ranges: 50–1455 kg m^−2^ s^−1^	3–35 °C, Evaporating 15–50 °C, Adiabatic 30–60 °C, Condensing		Qualities: 0–1	Void fractions: 0–1	QCV: 13 cases Optical: 6 cases Capacitance: 3 cases

* Abbreviation: MP: Multiport; SC: Smooth-circle tube; FT: Flattened; H: Horizontal; QCV: Quick-closing valve.

**Table 2 sensors-22-03511-t002:** Summary of experimental conditions.

Variable	Conditions
Mass flux, G (kg m^−2^ s^−1^)	100, 250, 400
Quality, x	0.025–0.9
Saturation temperature, Tsat (°C)	25
Saturation pressure, Psat (MPa)	1.6986
Refrigerant	R32

**Table 3 sensors-22-03511-t003:** Single-phase validation results.

	Capacitance Value (fF)	
Phase	FEM Analysis	Experiment
Gas	91.401	90.522
Liquid	149.88	149.295

**Table 4 sensors-22-03511-t004:** Discrepancy between the QCV technique and the capacitance method with the proposed calibration technique.

	*E_A_ **		*R* ^2^	
	Non-Calibrated	Calibrated	Non-Calibrated	Calibrated
Slug	30.4%	7.8%	0.287	0.966
Intermittent	25.33%	2.24%	−0.198	0.986
Stratified	9.02%	0.63%	0.499	0.997
Annular	4.54%	1.36%	−1.307	0.776
All	19.53%	2.99%	0.711	0.994

* *E_A_*: average relative error.

## Data Availability

Not applicable.

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
