# Peer review of "Optimizing Calibration for a Capacitance-Based Void Fraction Sensor with Asymmetric Electrodes under Horizontal Flow in a Smoothed Circular Macro-Tube"

_sensors, 2022, doi:10.3390/s22093511_

Round 1
Reviewer 1 Report
The paper presents an optimal calibration method for void fraction, which is based on the flow pattern obtained from the capacitive sensor. The proposed calibration method was validated against the QCV method. This paper may be reconsidered after revision.
Q1
The references cited in the article are irregular.
Q2
The abstract lacks a description of the study methodology, and there should be more description of the calibration method specifics.
Q3
“1. Introduction”: “Table 1”listed void fraction measurement results lack qualitative and quantitative analysis, and a general summary should be made in Table 1.
Q4
Line 125: “…prevent oil mixing and pulsation.” need more explanation.
Q5
Line 181: “…reduction of the void fraction data”,why is an important parameter for reducing the void fraction data.
Q6
Lines 54-58 of the paper describe the limitations of QCV, then a description and analysis of this part should be added to Section “3.1 Measurement apparatus for the QCV technique”.
Q7
The description of the capacitive sensor in Section “3.3 Measurement apparatus for the capacitance technique” states that it is installed in the center of the test section, but the “1 introduction” mentions the use of "…validate a capacitance method for void fraction measurement by applying a capacitance sensor [31] with low sensitivity to measure flow patterns", please explain clearly; if there is a physical (real) picture, please put it in the paper.
Q8
Line 246, "…the flow pattern type was separated into two general groups" is three flow patterns according to Figure 2, please explain this part.
Q9
Line 255, "…Note that the y axis in Figure 8 shows the normalized capacitance (Cn)." Please identify Figure 7 or Figure 8
Author Response
Dear reviewer
The authors are sending the answer sheet to reviewer's comments and revised manuscript.
Although all comments were general questions, the authors also agree that all comments are good and necessary to enhance the quality of manuscript.
All comments were reflected to the revised paper, doing the best.
We appreciate that you handle with our manuscript.
Best Regards.

Reviewer 2 Report
Review comments on: Optimizing calibration for capacitance-based void fraction sensor with asymmetric electrodes under horizontal flow in a smoothed circular macro tube
The authors present a very good study on the non-linear calibration of a capacitance sensor with asymmetric electrodes. Furthermore the calibration was done for different flow regimes which is not usually seen in the literature, hence it is well appreciated by the reviewer. It is the reviewer’s opinion that the paper may be accepted after the following comments are addressed:
- Flow direction should be indicated in fig. 3. I suppose the bypass to the QCV is at the end of the pipe not from the inlet section. This is important otherwise some readers may assume the flow direction in both pipes is the same rather than in feedback or return fashion.
- A clear reason should be given why the intermittent and slug flow regimes gave relatively higher deviations from the QCV void fractions.
- Error bars should be added to your plots of experimental data especially those obtained by your QCV, knowing fully well that QCV can have high uncertainty at some conditions.
- The flow development before any measurements are taken should be around 50D. Has this been achieved before your capacitance sensor measurements? Could it be that the figure is not properly drawn? Giving the dimensions and distances in the figure will help the reader and in your explanations.
- Some samples of your capacitance sensor time histories should be given for the different flow regimes.
Author Response

(The authors gave the same response as above.)

Reviewer 3 Report
The authors developed a capacitance method for void fraction measurement by applying a capacitance sensor with low sensitivity to measure flow patterns. The present study provided a potential alternative for the QCV measurement method. In general, the paper is well written and organized, and the results are satisfying. My concerns with this work is as follows:
- The FEM analysis was employed to analyze the C-α curves for each representative flow pattern. So how to perform the FEM analysis? More details are suggested to be provided though the paper itself is long enough.
- It is still not very clear for us to understand the calculation processes of the void fraction for the void fraction measurement methodology of the capacitance sensor calibration. Is it possible to provide a flow chart to show this process?
- The present method was only validated using the experimental data by the authors themself. We know that there are lots of studies concerning the void fraction measurement for refrigerants. If the present method could becompared to other measurement techniques under the their experimental conditions?
- In Line 134, Page 4 of 18, the authors state that “......an accuracy of ±0.05 %.”, the accuracy is incredibly high, please check it.
Author Response

(The authors gave the same response as above.)

Round 2
Reviewer 1 Report
The author has a good answer to the questions raised, and the author gives a complete explanation of the method of measuring void fraction using capacitive sensors. I think this article can be accepted if the following issues are resolved:
The author has made appropriate supplements to "Table 1", but there should be an analysis and description of the gas content measurement methods listed in Table 1, summarizing the current research status. It is clear that the existence of the “Table 1” would be meaningless if it were merely a count of what is listed in the table.
Author Response
The authors are sending the answer sheet to reviewer's comments and second version of revised manuscript.
Although all comments were general questions, the authors also agree that all comments are good and necessary to enhance the quality of manuscript.
All comments were reflected to the revised paper, doing the best.
We appreciate that you handle with our manuscript.
Best Regards.

Reviewer 2 Report
The authors revised the manuscript but did not address all comments satisfactorily Specifically,
comment 4: You only revised the rig schematic but did not address the flow development with respect to placement of the capacitance sensor such that it is without entrance effects. Flow development is is important in two-phase flow, and the authors will improve the usefulness of paper to its audience by commenting on this.
comment 5: The PDFs given are no substitutes for the time histories of the void fraction. Please provide representative time series and map them to the images.
Additionally, please improve the quality of your images especially figures 2, 6, 7, and 9.
Author Response

(The authors gave the same response as above.)

Reviewer 3 Report
The authors have well responded my concerns and made necessary modifications. I think the manuscript can be accepted now.
Author Response
The authors are sending the answer sheet to reviewer's comments and second version of revised manuscript.
Although all comments from other reviewers were general questions, the authors also agree that all comments are good and necessary to enhance the quality of manuscript.
All comments were reflected to the revised paper, doing the best.
We appreciate that you handle with our manuscript.
Best Regards.
